# Loss of Control over Eating, Inhibitory Control, and Reward Sensitivity in Children and Adolescents: A Systematic Review

**DOI:** 10.3390/nu15122673

**Published:** 2023-06-08

**Authors:** Sofia Marques Ramalho, Eva Conceição, Ana Cristina Tavares, Ana Luísa Freitas, Bárbara César Machado, Sónia Gonçalves

**Affiliations:** 1Centro de Investigação em Psicologia para o Desenvolvimento (CIPD), 4100-346 Porto, Portugal; 2Instituto de Psicologia e de Ciências da Educação, Universidade Lusíada (Porto), 4100-348 Porto, Portugal; 3Psychotherapy and Psychopathology Research Unit, Psychology Research Centre, School of Psychology, University of Minho, 4704-553 Braga, Portugal; econceicao@psi.uminho.pt (E.C.); anacvtavares@outlook.pt (A.C.T.); a93662@alunos.uminho.pt (A.L.F.); sgoncalves@psi.uminho.pt (S.G.); 4Research Centre for Human Development (CEDH), Faculdade de Educação e Psicologia, Portuguese Catholic University, 4169-005 Porto, Portugal; bcmachado@ucp.pt

**Keywords:** systematic literature review, executive functions, loss of control over eating, adolescents, children

## Abstract

Overview: In recent years, there has been increasing clinical and empirical interest in the concept of pediatric loss of control over eating, particularly about its link with the executive functions related to the concept of impulsivity, such as inhibitory control and reward sensitivity. However, there has yet to be a comprehensive literature synthesis about the associations between these variables. A comprehensive literature synthesis would help identify future research directions to advance the field in this area. Therefore, this systematic review aimed to synthesize evidence concerning the associations between loss of control over eating, inhibitory control, and reward sensitivity in children and adolescents. Methods: The systematic review was conducted according to the guidelines proposed by PRISMA in Web of Science, Scopus, PubMed, and PsycINFO. The Quality Assessment Tool for Observational Cohort and Cross-Sectional Studies was used to assess the risk of bias. Results: Twelve studies met the selection criteria and were included in the final review. Overall, methodological heterogeneity, variability in assessment methods, and the age of participants make it difficult to draw general conclusions. Nevertheless, most studies with community samples of adolescents indicate that inhibitory control difficulties are linked to the concept of loss of control eating. The presence of obesity seems to be associated with inhibitory control difficulties, regardless of the presence of loss of control eating. Studies on reward sensitivity are scarcer. However, it has been suggested that higher reward sensitivity is related to loss of control eating behaviors in young people, particularly binge eating. Conclusions: The literature on the link between loss of control eating and trait-level facets of impulsivity (low inhibitory control and higher reward sensitivity) among young people remains limited, and more studies on children are needed. Findings from this review may make healthcare professionals more aware of the potential clinical importance of targeting the trait-level facets of impulsivity and help to inform existing and future weight-loss/maintenance interventions in childhood and adolescence.

## 1. Introduction

Pediatric Loss of control over eating (LOC-eating) can be characterized by the subjective perception of being compelled or unable to control the amount and type of food consumed among young people [1]. The experience of LOC-eating can be associated with different eating episodes regardless of the amount of food consumed [2]. LOC-eating often results in subjective distress and/or eating beyond the point of satiety being considered an obesogenic eating behavior frequent in youth with overweight/obesity [3]. When present in childhood, it tends to persist in up to 50% of children and young people [1]. LOC-eating prospectively predicts excess weight gain and is related to several adverse developmental outcomes and psychosocial impairments such as emotional stress, depressive symptomatology, weight-based teasing, dieting, and cognitive functioning difficulties [1,4,5,6].

The term “executive functions” relates to a multidimensional construct that encompasses several complex cognitive processes. Examples of these include decision-making, planning, attention, problem-solving, inhibitory control, working memory, cognitive flexibility, etc. [7,8,9,10,11]. Inhibitory control and reward sensitivity are two executive functions that fall under the broader construct of impulsivity—a multi-dimensional construct describing a predisposition to engage in behaviors with reduced planning and seeking immediate reward despite the long-term consequences [12]. The construct of inhibitory control involves the ability to overcome an internal predisposition or external attraction by controlling attention, behavior, thoughts, and/or emotions to accomplish what is most appropriate [12]. It interferes with goal-directed behavior [13,14]. On the other hand, reward sensitivity refers to trait-level reactivity and responsivity to rewarding stimuli, including motivation to seek out rewards and tendencies to engage in approach behaviors [13,15]. 

The study of these two executive functions is significant since they are specifically associated with the onset and maintenance of obesity and interact to promote problematic eating behaviors, such as LOC-eating in young people [16,17,18,19]. Nevertheless, there is a relative scarcity of research and mixed conclusions about the link between these two constructs and LOC-eating in young people [20,21,22], and an examination of the existing literature may provide novel conclusions and directions for future studies. 

To our knowledge, this is the first study that systematically reviews the scientific literature regarding the associations between LOC-eating and executive functions, inhibitory control, and reward sensitivity in children and adolescents across the weight spectrum. Its main goal was to provide a comprehensive assessment of the current empirical evidence and identify the research gaps related to this topic. This work specifically assesses whether the relationship between LOC-eating, inhibitory control, and/or reward sensitivity varies by weight status (healthy weight, overweight, obesity) and summarizes information on the measures/procedures used in the literature to assess inhibitory control, reward sensitivity, and LOC-eating in this particular age group (Figure 1). Findings from this review could contribute to informing treatment development and alert healthcare professionals of the clinical importance of targeting these executive functions and LOC-eating in their interventions to prevent and treat overweight/obesity in childhood and adolescence. 

## 2. Methods

This systematic review followed the Preferred Reporting Items for Systematic Reviews and Meta-Analyses (PRISMA) statement [23].

### 2.1. Search Strategy 

A search was conducted up to February 2021 on the following databases: Web of Science, Scopus, PubMed, and PsycINFO. The following terms were searched: adolescence, adolescents, child, children, youth, executive function, inhibition, inhibitory control, reward, reward sensitivity, impulsivity, loss of control over eating, loss of control eating, uncontrolled eating, dysregulated eating, and binge eating. Titles and abstracts of the studies identified through the keyword search were screened against the study selection criteria. Eligible articles were retrieved, and their full texts were assessed. Two co-authors independently performed title and abstract screening. Discrepancies were resolved through discussion under the participation of a third co-author. The search algorithm strategy is provided in the Appendix A.

### 2.2. Study Selection Criteria

Studies that met all of the following criteria were included in the review: (1)study designs—experimental (e.g., randomized controlled trials or pre-post-studies) and observational studies (e.g., longitudinal or cross-sectional studies) with humans; (2) study subjects—children and/or adolescents with ages ranging from 7 to 18 years old (the rationale behind selecting this age range is grounded in the reading ability and in available self-report measures in the literature to assess eating behavior in infancy since we want to exclude papers using parent proxy-report data); (3) outcomes—Problematic eating behaviors associated with LOC-eating (e.g., binge eating; uncontrolled eating, LOC-eating) and at least assess one of the following executive functions: inhibitory control and reward sensitivity; (4) article type—peer-reviewed journal publications; (5) time window of search—from January 2000 to February 2021; and (6) language—articles written in English or Portuguese. 

Duplicate studies and studies that met any of the following criteria were excluded from the review: (1) Studies that examined either samples with bulimia nervosa or anorexia nervosa since the focus of our work was on community samples and samples with overweight/obesity; (2) studies that include participants with a neurodevelopmental disorder (e.g., attention-deficit/hyperactivity disorder) that may influence their performance in executive functioning. If the study includes participants with and without these conditions, only results from participants without these conditions will be analyzed and described; (3) studies that only report an overall executive functioning score or do not provide it in the results, conclusion, and/or discussion sections information about the associations’ under study; and (4) book/book chapters, thesis, letters, editorials, study/review protocols, case reports, meta-analysis, or narrative or systematic review articles.

### 2.3. Data Extraction and Synthesis

A standardized data extraction form was used to collect the following information from each included study: (1) study identification (title; journal; authors; language; year of publication); (2) methodological characteristics (study design and aims; executive functions assessed; measures used to assess the LOC-eating and the inhibitory control and/or reward sensitivity; sample characteristics (sample size, sex, age, and BMI z-score); and (3) key outcomes/conclusions regarding the relationship between LOC-eating, inhibitory control, and reward sensitivity. Two co-authors independently conducted the data extraction. After the initial screening of the study title, abstract, and type (e.g., RCT, etc.), the full texts of potential studies were screened in order to determine eligibility for inclusion. Study exclusion motives were documented. Final results were compared and conflicts between reviewers were resolved through discussion with a third co-author. 

### 2.4. Assessment of Risk of Bias 

Two independent reviewers used the *Quality Assessment Tool for Observational Cohort and Cross-Sectional Studies—National Institutes of Health* [24] to assess the risk of bias in the cross-sectional/observational studies included in this systematic review and the strength of their scientific evidence (not to determine the study inclusion). Disagreements were resolved by discussion and consultation with a third reviewer. For each of the 14 criteria, a score of 1 was assigned if “yes” was the response, whereas a score of 0 was assigned otherwise. A study-specific global score, ranging from 0 to 14, was calculated by summing up scores across all criteria. Quality was rated as poor (0–4 “yes” answers out of 14 questions), fair (5–10 “yes” answers out of 14 questions), or good (11–14 “yes” answers out of 14 questions).

## 3. Results

### 3.1. Study Selection

Figure 2 shows the PRISMA study selection flowchart. A total of 919 articles were identified through the databases (367 in Web of Science, 357 in Scopus, 95 in PubMed, and 100 in PsycINFO). No additional sources were considered. After removing duplicates, 473 articles underwent title and abstract screening, of which 454 were excluded according to the study selection criteria. This selection process resulted in the full-text examination of 19 articles, of which 7 articles were excluded for the following reasons: (a) they did not present data regarding the associations between LOC-eating and inhibitory control and/or reward sensitivity in the results, conclusion, or discussion sections (*n* = 4); (b) they presented a global measure of eating behavior psychopathology, but did not explicitly evaluate LOC-eating (*n* = 1); (c) they assessed general executive functioning rather than inhibitory control/reward sensitivity scores (*n* = 1); and (d) the results regarding the relationship between LOC-eating and inhibitory control did not contain any data for the sample without attention-deficit/hyperactivity disorder. Therefore, a total of 12 articles were included in this review [18,25,26,27,28,29,30,31,32,33,34,35].

### 3.2. Study Characteristics

Table 1 summarizes the characteristics of the twelve studies included in this systematic review regarding their sample size/characteristics, measures used to assess inhibitory control, reward sensitivity, LOC-eating, and main conclusions. Despite the inclusion of a study with a prospective design, the remaining studies followed a cross-sectional/observational study design. The age of participants across studies ranged from 8 to 18 years. One study was conducted exclusively with a sample of female adolescents [31], and the remaining studies involved participants from both sexes. The percentage of females ranged from 52% to 81.8%. The study sample size varied substantially across studies, from 40 to 4803 participants. Six studies focused exclusively on a community sample of adolescents [25,27,29,30,31,35], three studies focused on the comparison between children/adolescents with overweight/obesity, healthy weight, and/or binge-eating disorder [26,28,34], and one study focused exclusively on children and adolescents with overweight/obesity [18]. The research measures used in these studies include mostly behavioral tasks (one during fMRI) and self-report measures [26].

### 3.3. Quality Assessment

Table 2 describes the criterion-specific report of the studies identified as relevant for this review, considering the National Institutes of Health (NIH) Quality Assessment Tool for Observational Cohort and Cross-Sectional Studies [24]. Of the twelve studies included in the review, ten were classified as “fair” (5–10 out of 14 questions), and two were rated as “good” (11–14 out of 14 questions). All studies clearly defined the research questions and study population. The participant rate was representative of the target population, and all of them indicated eligibility criteria and how participants were recruited, clearly defining the independent and dependent variables under study. All of the studies took into account the potential impact of confounding variables in their statistical analysis. Yet, except for the studies conducted by Van Malderen et al. [35] and Goldschmidt et al. [34], the remaining ten studies did not provide a sample size justification.

### 3.4. Measures of Loss of Control Eating, Inhibitory Control, and Reward Sensitivity

Table 1 summarizes the measures used to assess inhibitory control, reward sensitivity, and LOC-eating in children and adolescents in the included studies. Five studies used the Children’s Eating Disorder Examination (ChEDE)—a semi-structured interviewer-based instrument to evaluate the presence of LOC-eating and rule out eating disorders [36]. Self-report questionnaires were also applied, namely the following: (1) “Loss of Control Over Eating Scales—Brief [LOCES-B]” [37], (2) “Children’s Eating Disorder Examination Questionnaire [ChEDE-Q]” [36,38], (3) “Three Factor Eating Questionnaire R-18 (TFEQ-R18)”—subscale: “uncontrolled eating” [39], (4) “Youth Risk Behavior Surveillance System Questionnaire” [40], and the “Eating Disorder Diagnostic Scale—Binge Eating Subscale” [41].

Regarding inhibitory control, six studies applied just behavioral tasks and four applied just self-report questionnaires. The behavioral tasks used to assess inhibitory control were as follows: “Go/NoGo Task adapted” [42,43,44], “Stop Signal Task” [33,45], “Opposite Worlds Task” [46], “Groton Maze Timed Chase Task” [18], “Flanker Test” [47], “Dimensional Change Card Sort Test” [48], and the “Stroop Color-Word Interference Test” [49]. On the other hand, all the studies used the same self-report measure to evaluate inhibitory control—“*Behavior Rating Inventory of Executive Function* (BRIEF)—Inhibition Subscale” [50]. One study applied a behavioral task (“Go/NoGo Task”) and a self-report measure simultaneously to assess inhibitory control in adolescents [30].

Five studies assessed reward sensitivity. The behavioral tasks used included the “Door opening task” in two studies [26,32] and the “Delay of Gratification Task” in one study [32]. Self-report measures included the “Sensitivity to Punishment and Sensitivity to Reward Questionnaire for Children” [51] and the “Behavioral Activation Scale” [52] in two studies [18,25] to assess reward sensitivity. One study applied the “Event-related reward-guessing task during an fMRI scan” [31].

### 3.5. Loss of Control Eating and Inhibitory Control

The relationship between LOC-eating and inhibitory control was explored in 10 out of the 12 studies included in the present review, resulting in mixed conclusions [18,25,26,27,28,29,30,33,34,35]. When considering objective behavioral task-based measures of inhibitory control, the conclusions regarding the link between LOC-eating and inhibitory control are incongruent and mostly focused on adolescents (vs. children). Inhibitory control impairments were strongly associated with binge eating behavior in a community sample of adolescents [27], and adolescents experiencing LOC-eating and obesity also showed poorer inhibitory control when compared to adolescents of a normal weight [28]. On the other hand, in a community sample considering children and adolescents, the performance on a behavioral task assessing inhibitory control (contrarily to the self-report measure) did not significantly predict overall disordered eating [30]. In another prospective study, greater inhibitory control at 10 years was linked with a decreased risk for disordered eating (in which binge eating was assessed) at age 14 [33].

In samples that included exclusively children with overweight/obesity, no differences in inhibitory control were found between children with obesity and binge eating episodes and children with obesity without binge eating episodes [26]. Additionally, there were no differences between the normal weight group, the group considered overweight, and the group considered overweight and prone to concomitant LOC-eating in terms of performance in behavioral tasks assessing inhibitory control [34].

### 3.6. Loss of Control in Eating and Reward Sensitivity 

The relationship between LOC-eating and reward sensitivity was explored in five of the studies included in this review [18,25,26,31,32]. Of these, three studies reported no associations between LOC-eating and reward sensitivity in children and adolescents (with and without overweight/obesity) [18,25,32]. Two studies found a relationship between the presence of LOC-eating and higher reward sensitivity. Specifically, Nederkoorn and colleagues [26] showed that children with obesity under residential treatment were more sensitive to rewards than children of a healthy weight and that those who reported binge eating episodes before treatment appeared to be more sensitive to rewards than the children with obesity without binge eating episodes. Additionally, results from a neuroimaging study of community samples of female adolescents suggested that higher reward sensitivity was correlated to greater activation of the ventromedial prefrontal cortex (the medial portion of the prefrontal cortex implicated in a variety of functions such as decision-making, social cognition, emotional processing, and memory) which, in turn, was related to binge-eating severity (where LOC-eating is present) [31].

## 4. Discussion

The main aim of this systematic review was to understand the relationship between LOC-eating, inhibitory control, and reward sensitivity in children and adolescents, synthesizing and assessing the existing literature that has accumulated over the past 20 years. Twelve studies met the selection criteria and were included in the review [18,25,26,27,28,29,30,31,32,33,34,35]. Overall, research on this topic remains limited, and existing studies primarily focus on adolescents. Moreover, most of the existing studies adopt a cross-sectional design, precluding causal assumptions.

Mixed results, methodological heterogeneity, greater variability in assessment methods, and participants’ age/conditions make it difficult to draw general conclusions, and more research is needed. Yet, it seems that the majority of the studies indicate that low inhibitory control and higher reward sensitivity are linked to LOC-eating and obesity [27,28,30]. None of the studies included in this review presented a high risk of bias. When compared with healthy weight controls, inhibitory control difficulties were associated with the obesity weight status per se. Overall, these results are in line with the existing literature on adults [53], showing that adults with obesity exhibit deficits in inhibitory control [16,17,54,55]. This may indicate the existence of neurocognitive/cognitive processing endophenotypes in pediatric obesity that could have clinical implications for treatment outcomes.

Nevertheless, the existing literature presents heterogenic results that may be explained by divergent assessment methods of inhibitory control, reward sensitivity, and LOC-eating in children and adolescents (behavioral tasks vs. self-report questionnaires) [25,56]. Regarding inhibitory control self-report measures, most studies applied the Inhibition Subscale of the Behavioral Rating Inventory of Executive Function [50]. However, across the literature, there was a high discrepancy on the behavioral tasks selected to assess this construct. Distinctive behavioral tasks may measure different dimensions of the same construct [29]. This rationale is also applicable to reward sensitivity, where multiple behavioral tasks were used to assess the same construct. Self-report questionnaires probably assess stable and dependent aspects similar to self-regulation “traits” (typical performance) [57,58]. On the other hand, measures based on behavioral tasks are probably better suited to assess aspects related to the self-regulation “state” (maximal performance), as they are vulnerable to temporal fluctuations, contributing to the ambivalent results shown in the present study [57].

Additionally, self-regulation and executive functions are not fully developed until adulthood. In particular, inhibitory control shows a maturation peak during adolescence [59], and the adverse outcomes of low inhibitory control in adolescence on eating behavior may only emerge in adulthood [25]. Most studies included in this review do not consider the executive functions’ developmental trajectories and their maturation status in the interpretation of their results. That might help to explain why some studies representing different developmental stages did not find a relationship between LOC-eating and inhibitory control. 

### 4.1. Limitations

The selection of studies for this systematic literature review followed the guidelines proposed by PRISMA to avoid publication bias and ensure the quality of the selected studies. To reduce the risk of bias, a quality assessment of the articles included in the review was conducted. Nevertheless, significant limitations and methodological issues have emerged. Namely, (1) this systematic review found just three studies on the interaction between weight status, reward sensitivity/inhibitory control, and LOC-eating, preventing new concrete conclusions on this topic; (2) unpublished data and data in other languages besides English or Portuguese were not included; and (3) the definition of circumscribed key-search terms to find some homogeneity in the available data could potentially lead to the exclusion of some studies relevant to this topic discussion. Moreover, although no articles with a high risk of bias were included in the review, an evaluation of the overall quality of the articles revealed that 10 of the articles included in the review were of fair quality, and the results of these studies should be interpreted carefully. 

The studies included in the review also pointed out some limitations that should be taken into consideration. For instance, though they do not provide sample size justification, several studies list a small sample size as a potential limitation in the interpretation of their results [18,25,28,32,35]. The use of only one type of executive function measurement and the use of self-report measures to assess inhibitory control is noted by some authors as a study limitation [29,35]. Finally, the results obtained in studies involving community samples of adolescents may not be generalizable to clinical samples.

### 4.2. Recommendations for Future Studies

The results of this systematic review highlight key areas to address in future research. First, there is increasing evidence for good psychometric properties of the Loss of Control Over Eating Scale (LOCES-B) when used in early adolescents [60], but the psychometric adaptation of this instrument to children is needed to expand future research on this specific construct and enable the study of developmental trajectories. The use of a clinical interview, such as the Eating Disorder Examination [61], to diagnose the presence of eating disorders in which LOC-eating is present (e.g., binge-eating disorder) can be time-consuming and requires experts with extensive clinical training [62,63]. This may limit advancements in this research field. Finally, using a multi-method assessment (self-report plus behavioral tasks) of inhibitory control and reward sensitivity could help to reduce bias in the results related to the evaluation of distinctive dimensions of the same construct [29,35].

Secondly, technology may also facilitate a more ecologically valid assessment of these variables through Ecological Momentary Assessment (EMA). Since it facilitates the intensive longitudinal assessment of target behaviors, it could ultimately help to find new insights into the prediction and modeling of the casual associations between disordered eating, weight, and executive functioning. Considering the absence of longitudinal studies evaluating the causality and directionality of the relationship between LOC-eating, inhibitory control, and reward sensitivity, it would be beneficial for future research to fill this gap in the literature [18,25,26,27,29,30,32,34]. Finally, studies focused on mood assessment could be relevant, as mood seems to play a relevant role in LOC-eating and its relationship with executive functions [35].

Executive functioning (which influences cognitions, emotions, and behaviors linked to obesity) may be a significant yet under-emphasized factor in informing current and future pediatric obesity interventions. It may underlie both obesity and LOC-eating behaviors across the age spectrum, but there is a relative paucity of research on children and adolescents with both conditions. In addition, a small amount of research has explored how low inhibitory control and higher reward sensitivity are related to behaviors that can promote weight gain in children and adolescents [18].

Overall, increasing our knowledge of the interaction between inhibitory control and reward sensitivity and whether they are associated with obesity and obesity-related behaviors in children and adolescents could help to develop personalized interventions to improve treatment outcomes and reduce the dropout rate in weight-loss interventions. Furthermore, intervening earlier in a child’s development may lead to greater success in reducing weight gain and preventing adult obesity.

## 5. Conclusions

This study systematically reviewed the scientific literature about the relationship between LOC-eating, inhibitory control, and reward sensitivity in children and adolescents across the weight spectrum. Overall, this is a topic of growing interest, as evidenced by the fact that 10 of the 12 studies included in this review were published in the last five years. Methodological heterogeneity and greater variability in the measurement of the key variables make it difficult to draw general conclusions. Nevertheless, despite conflicting findings, it seems to be the case that inhibitory control difficulties are linked to LOC-eating mostly in community samples of adolescents. The presence of obesity seems to be independently associated with inhibitory control difficulties. Studies on reward sensitivity are scarce and mixed, with most of the studies (three out of five) showing no association. Yet, two studies found that higher reward sensitivity is related to LOC-eating behaviors and obesity in children, particularly when binge eating is present.

In conclusion, more studies on children are needed, and future research should consider longitudinal designs and the implementation of ecological momentary assessment (EMA) protocols in neurocognitive assessments. Prevention and treatment interventions in pediatric obesity should consider the shared risk factors between pediatric obesity and eating psychopathology [64], and the potentialities of early executive function training and intervention in problematic eating behaviors, such as LOC-eating, to stabilize or prevent weight gain [65].

## Figures and Tables

**Figure 1 nutrients-15-02673-f001:**
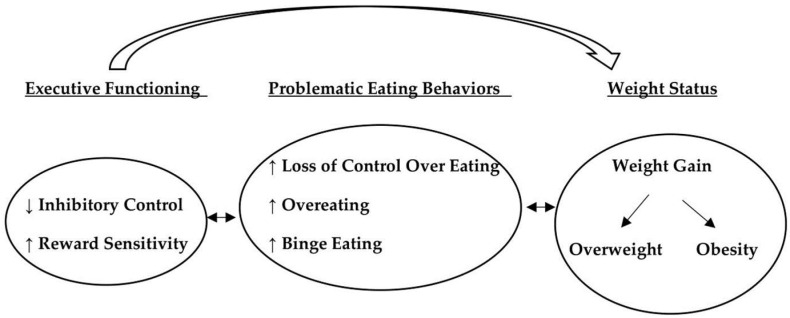
Hypothesized associations among Executive Functions, Problematic Eating Behaviors and Weight Status. Note: ↑ arrow indicates increase. ↓ arrow indicates decrease. Bidirectional arrows indicate reciprocal influences between Problematic Eating Behaviors and other variables.

**Figure 2 nutrients-15-02673-f002:**
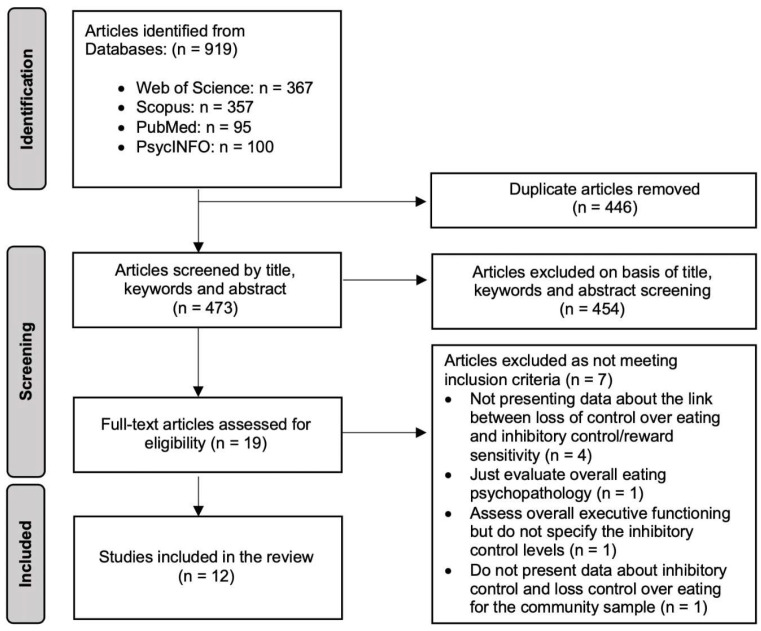
PRISMA flowchart illustrating the study selection process.

**Table 1 nutrients-15-02673-t001:** Characteristics of the studies included in the review.

Author/Year	Sample Characteristics				Measures			Key Findings
	Population	Sample Size	Age Range (*M*, *SD*)	Female %	Inhibitory Control	Reward Sensitivity	Loss of Control Eating	
Van Malderen et al. (2021) [35]	Adolescents Community Sample	*N* = 50	10–18 years (14.22, 2.58)	76%	“Behavior Rating Inventory of Executive Function” (BRIEF)—“inhibition subscale”	_________	Loss of Control over Eating Scale-Brief (LOCES-B)	– In the neutral condition (no negative mood induction), inhibition subscale scores were not significantly associated with loss of control eating. – In the condition where negative mood was induced, low inhibitory control was positively associated with loss of control eating.
Nelson et al. (2020) [30]	Adolescents Community Sample	*N* = 208	14–16 years (14.5, 0.75)	52%	“GO/No-GO Task” “Behavioral rating inventory of executive function, second edition” (BRIEF-2)—“inhibit subscale”	__________	“Three Factor Eating Questionnaire R-18” (TFEQ-R18)—scale: “uncontrolled eating”	– The inhibit subscale significantly and uniquely predicted uncontrolled eating (more inhibitory control difficulties associated with greater loss of control eating). – Performance on the Go/No-Go task was not a significant predictor of uncontrolled eating.
Schaumberg et al. (2020) [33]	Children and adolescents Community Sample	*N* = 4803	8; 10; 14; 16; 18 years	_________	“Stop signal task” “Opposite worlds task”	__________	“Youth Risk Behaviour Surveillance System questionnaire”	– Measures of inhibition did not show a relationship with eating disorder symptoms throughout adolescence. – Higher scores on the Stop Signal Task (reflecting greater inhibitory control) were associated with decreased risk of binge eating disorder at age 14. – No consistent relationships were found between inhibition difficulties and binge eating.
Van Malderen et al. (2019) [29]	Adolescents Community Sample	*N* = 301	10–17 years (13.46, 1.99)	67.2%	“Behavior Rating Inventory of Executive Function (BRIEF)”—“inhibition subscale”	_________	“Children’s Eating Disorder Examination Questionnaire” (ChEDE-Q)	– The results provided no evidence for the unique effects of inhibitory control in predicting binge eating.
Munsch et al. (2019) [32]	Children Community Sample	*N* = 100	8–13 years	57%	___________	“Door opening task” (DOT) “Delay of gratification task” (DoGT)	“Eating Disorder Examination adapted for Children” (ChEDE)	– No significant differences were found in the level of reward sensitivity between children with and without loss of control eating.
Goldschmidt et al. (2019) [19]	Children and adolescents with overweight/ obesity	*N* = 40	8–14 years (11.15, 1.89)	55%	“Groton Maze Timed Chase Task” (GMTCT)	“Sensitivity to Punishment and Sensitivity to Reward Questionnaire for children” (SPSRQ-C)	“Child Eating Disorder Examination 12.0” (ChEDE)	– Higher numbers of errors on a behavioral measure of visuo-motor processing (reflecting less inhibitory control) were related to a greater overall severity of loss of control eating. – The results suggest that impulsivity and inhibitory control may influence an individual’s tendency to engage in disordered eating behaviors. – No effect was found for reward sensitivity on loss of control in eating.
Van Malderen et al. (2018) [25]	Adolescents Community Sample	*N* = 124	10–17 years (14, 1.90)	65.3%	“Behavior Rating Inventory of Executive Function” (BRIEF)—“inhibition subscale” “Behavioral Inhibition Scale” (BIS)	“Behavioral Activation Scale” (BAS)	“Children’s Eating Disorder Examination” (Ch-EDE)	– There were no significant differences between the group that experienced loss of control in eating and the group that did not, in the subscale measuring inhibition. – The reward sensitivity scale did not play a significant moderating role in the relationship between the inhibition subscale and loss of control eating.
Goldschmidt et al. (2018) [34]	Children of a healthy weight, overweight children, and obese children	*N* = 75	9–12 years (10.5, 1.1)	58.7%	“Flanker Test” “Dimensional Change Card Sort Test”	_________	“Child Eating Disorder Examination 12.0” (Child EDE)	– The results of behavioral measures (neurocognitive tasks) that assessed inhibitory control did not differ between the groups (healthy weight control group, overweight control group, and overweight group with loss of control eating).
Bodell et al. (2018) [31]	Female adolescents’ Community Samples	*N* = 122	16; 18 years	100%	_____________	“Event-related reward-guessing task during an fMRI scan”	“The Eating Attitudes Test” (EAT)	– Compared to participants without binge eating disorder, participants prone to binge eating episodes showed a greater response from the caudate nucleus in a task with a monetary reward. – The results showed an association between greater activation of the ventro medial prefrontal cortex during reward receipt and the severity of binge eating. – No prospective association was found between ventromedial prefrontal cortex activation and binge eating.
Kittel et al. (2017) [28]	Adolescents with Binge Eating Disorders + Obesity (BED), with Obesity (O), and with healthy weight (*N*)	*N* (BED) = 22 *N* (O) = 22 *N* (*N*) = 22	BED = (14.91, 2.22) O = (14.82, 2.63) *N* = (15.23, 2.39)	BED = 81.8% O = 81.8% *N* = 81.8%	“Stroop Color–Word Interference Test”	___________	“Eating Disorder Examination” (EDE)	– In the task assessing inhibitory control (reflecting lower inhibitory control), compared to adolescents with healthy weight, the group of adolescents with binge eating disorder and obesity and the group of adolescents with just obesity showed no differences in performance.
Ames et al. (2014) [27]	Adolescents’ Community Samples	*N* = 198	14–17 years (15.84, 0.94)	56.1%	“Go/No-Go task” (food cued Go/NoGo and generic Go/No-Go)	___________	“Eating Disorder Diagnostic Scale”—“Binge Eating Subscale”	– Greater inhibition difficulties, as assessed by the Go/NoGo tasks, were significantly associated with binge eating in the girls’ sample. – Inhibition difficulties on the generic Go/NoGo task were significantly associated with binge eating behavior and BMI percentile.
Nederkoorn et al. (2006) [26]	Children with obesity (O); with obesity and binge eating (O+C); with obesity without binge eating (O−C); with healthy weight (*N*)	*N* (O) = 32 *N* (O + C) = 15 *N* (O − C) = 15 *N* (*N*) = 31	O = 12–15 years (13.7) O + C = 12–15 years (13.7) O − C = 12–15 years (13.9) *N* = 13–15 years (13.7)	O = 19/32 O + C = 10/15 O−C = 9/15 *N* = 19/31	“Stop Signal Task”	“The opening door task”	“Eating Disorder Examination—Questionnaire” (EDE-Q)	– Children with obesity and who are prone to binge eating episodes and children with obesity but do not struggle with binge eating did not differ in inhibitory control. – Children with obesity and who are prone to binge eating were more sensitive to reward than children with obesity but do not struggle with binge eating.

Note: In Munsch et al. (2019) [32], only the results of participants without attention deficit/hyperactivity disorder were analyzed and reported.

**Table 2 nutrients-15-02673-t002:** Study quality assessment.

Autor/Year	Q1—Research Question	QQ2—Study Population	Q3—Participation Rate	Q4—Recruitment and Eligibility Criteria	Q5—Sample Size Justification	Q6—Exposure Assessed Prior to Outcome Measurement	Q7—Sufficient Timeframe to See an Effect	Q8—Different Exposure Levels of Interest	Q9—Exposure Measures and Assessment	Q10—Repeated Exposure Assessment	Q11—Outcome Measures	Q12—Blinding of Outcome Assessors	Q13—Follow-up Rate	Q14—Statistical Analyses	Total Score	Overall Article Quality
Van Malderen et al. (2021) [35]	Yes	Yes	Yes	Yes	Yes	No	No	N.A.	Yes	N.A.	Yes	N.R.	N.A.	Yes	8	Fair
Nelson et al. (2020) [30]	Yes	Yes	Yes	Yes	No	No	No	N.A.	Yes	N.A.	Yes	N.A.	N.A.	Yes	7	Fair
Schaumberg et al. (2020) [33]	Yes	Yes	Yes	Yes	No	Yes	Yes	N.A.	Yes	Yes	Yes	N.A.	Yes	Yes	11	Good
Van Malderen et al. (2019) [29]	Yes	Yes	Yes	Yes	No	No	No	N.A.	Yes	N.A.	Yes	N.A.	N.A.	Yes	7	Fair
Munsch et al. (2019) [32]	Yes	Yes	Yes	Yes	No	No	No	N.A.	Yes	N.A.	Yes	N.A.	N.A.	Yes	7	Fair
Goldschmidt et al. (2019) [18]	Yes	Yes	Yes	Yes	No	No	No	N.A.	Yes	N.A.	Yes	N.A.	N.A.	Yes	7	Fair
Van Malderen et al. (2018) [25]	Yes	Yes	Yes	Yes	No	No	No	N.A.	Yes	N.A.	Yes	N.A.	N.A.	Yes	7	Fair
Goldschmidt et al. (2018) [34]	Yes	Yes	Yes	Yes	Yes	No	No	N.A.	Yes	N.A.	Yes	N.A.	N.A.	Yes	8	Fair
Bodell et al. (2018) [31]	Yes	Yes	Yes	Yes	No	Yes	Yes	N.A.	Yes	Yes	Yes	N.A.	N.R.	Yes	10	Good
Kittel et al. (2017) [28]	Yes	Yes	Yes	Yes	No	No	No	N.A.	Yes	N.A.	Yes	N.A.	N.A.	Yes	7	Fair
Ames et al. (2014) [27]	Yes	Yes	Yes	Yes	No	No	No	N.A.	Yes	N.A.	Yes	N.A.	N.A.	Yes	7	Fair
Nederkoorn et al. (2006) [26]	Yes	Yes	Yes	Yes	No	No	No	N.A.	Yes	N.A.	Yes	N.A.	N.A.	Yes	7	Fair

Note: This study quality assessment was adopted from the National Institutes of Health (NIH) Quality Assessment Tool for Observational Cohort and Cross-Sectional Studies Quality was rated as poor (0–4 yes answers out of 14 questions), fair (5–10 yes answers out of 14 questions), or good (11–14 yes answers out of 14 questions); N.A.—Not applicable; N.R.: Not Reported.

## Data Availability

A data availability statement is not applicable because this study is based exclusively on published literature.

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
