# Peer review of "Loss of Control over Eating, Inhibitory Control, and Reward Sensitivity in Children and Adolescents: A Systematic Review"

_nutrients, 2023, doi:10.3390/nu15122673_

Round 1

Reviewer 1 Report

This paper reports the results of a well-conducted systematic review of studies on the association between children and adolescents' loss of control over eating and inhibitory control and reward sensitivity.  Strengths of this well-written paper include appropriate methods for reviewing the literature, the assessment of study quality, clear reporting of study findings, acknowledgement of study limitations, and clear directions for future research.  The  major limitation is that since so little research has been conducted in this area (only 12 relevant studies were located) this systematic review summarizes a rather small number of studies. I have only two minor suggestions for improving the manuscript.  First, the authors need to justify why they did not include studies of children under the age of 7 (i.e., they need to justify the age range reviewed) and second, they should clarify a minor inconsistency.  On line 163 they state that all of the studies identified for this review were cross-sectional in design, although on line 229 they state that one of the studies employed a prospective design.

Author Response

REVIEWER 1

This paper reports the results of a well-conducted systematic review of studies on the association between children and adolescents' loss of control over eating and inhibitory control and reward sensitivity.  Strengths of this well-written paper include appropriate methods for reviewing the literature, the assessment of study quality, clear reporting of study findings, acknowledgement of study limitations, and clear directions for future research.  The  major limitation is that since so little research has been conducted in this area (only 12 relevant studies were located) this systematic review summarizes a rather small number of studies. I have only two minor suggestions for improving the manuscript.  First, the authors need to justify why they did not include studies of children under the age of 7 (i.e., they need to justify the age range reviewed) and second, they should clarify a minor inconsistency.  On line 163 they state that all of the studies identified for this review were cross-sectional in design, although on line 229 they state that one of the studies employed a prospective design.

Thank you for your comments. We have changed the minor inconsistency and justified the age range.

Reviewer 2 Report

I appreciate the opportunity to review the article titled: oss of Control Over Eating, Inhibitory Control and Reward

Sensitivity in Children and Adolescents: A Systematic Review. About this work I have the following suggestions:

1.- Could the authors include, in the introduction, an image that graphically summarizes the possible relationships between the study variables? This would help readers, at a glance, to understand what is explained in a few paragraphs.

2.- Figure 1, in the results section, is mentioned in the text, but it is not included and could not be reviewed by this reviewer. This is essential for the article to be published as a systematic review.

3.- Is it possible to reduce the font size of the two tables so that they take up less space and are easier to read?

Author Response

REVIEWER 2

I appreciate the opportunity to review the article titled: oss of Control Over Eating, Inhibitory Control and Reward

Sensitivity in Children and Adolescents: A Systematic Review. About this work I have the following suggestions:

1.- Could the authors include, in the introduction, an image that graphically summarizes the possible relationships between the study variables? This would help readers, at a glance, to understand what is explained in a few paragraphs.

- Thank you for the suggestion. We have added a new figure in the introduction.

2.- Figure 1, in the results section, is mentioned in the text, but it is not included and could not be reviewed by this reviewer. This is essential for the article to be published as a systematic review.

- It was a submission process mistake. Figure 1 is now attached.

3.- Is it possible to reduce the font size of the two tables so that they take up less space and are easier to read?

- Thank you for the suggestion. This paper was already transformed according to the Nutrients guidelines. But if the Editor agrees the font size can be reduced.